https://doi.org/10.1038/s41467-019-13680-7　　**OPEN**

# The METLIN small molecule dataset for machine learning-based retention time prediction

Xavier Domingo-Almenara [1,4]*, Carlos Guijas[1], Elizabeth Billings [1], J. Rafael Montenegro-Burke [1], Winnie Uritboonthai[1], Aries E. Aisporna[1], Emily Chen[2], H. Paul Benton [1] & Gary Siuzdak [1,3]*

Machine learning has been extensively applied in small molecule analysis to predict a wide range of molecular properties and processes including mass spectrometry fragmentation or chromatographic retention time. However, current approaches for retention time prediction lack sufficient accuracy due to limited available experimental data. Here we introduce the METLIN small molecule retention time (SMRT) dataset, an experimentally acquired reverse-phase chromatography retention time dataset covering up to 80,038 small molecules. To demonstrate the utility of this dataset, we deployed a deep learning model for retention time prediction applied to small molecule annotation. Results showed that in 70% of the cases, the correct molecular identity was ranked among the top 3 candidates based on their predicted retention time. We anticipate that this dataset will enable the community to apply machine learning or first principles strategies to generate better models for retention time prediction.

[1] Scripps Center for Metabolomics, The Scripps Research Institute, La Jolla, CA, USA. [2] California Institute for Biomedical Research (Calibr), The Scripps Research Institute, La Jolla, CA, USA. [3] Department of Integrative Structural and Computational Biology, The Scripps Research Institute, La Jolla, CA, USA. [4] Present address: Centre for Omic Sciences, EURECAT – Technology Centre of Catalonia & Rovira i Virgili University joint unit, Reus, Catalonia, Spain. *email: xavier.domingoa@eurecat.org; siuzdak@scripps.edu

Machine learning (ML) has played and still plays a key role at different levels in fields as diverse as quantum mechanics, physical chemistry, biophysics or physiology[1]. In chemoinformatics, ML has been widely adopted in the design of quantitative structure–activity relationship (QSAR) models[2,3] aimed at predicting specific properties such as bioactivity[4], toxicity[5] or small molecule-protein binding affinity[6]. These models enable screening for molecules with specific properties and their development has been possible given the availability of public datasets. In that sense, datasets with a wide set of examples from which an ML model can learn are necessary to build accurate ML-based prediction models[7].

Liquid chromatography coupled to mass spectrometry (LC–MS) is routinely used in hundreds of laboratories for small molecule analysis[8]. In mass spectrometry-based small molecule analysis, ML has been applied in the prediction of collision-induced dissociation (CID) or electron ionization (EI) fragmentation spectra, known as in silico spectra[9–11] or in the design of spectra annotation software[12]. These ML-based resources enable scientists to identify and discover molecules that lack experimental spectra in databases, which constitute most of the known molecules[13,14]. Historically, the lack of publicly available datasets to train ML models has hampered the design of new algorithms for in silico spectra prediction or annotation. Generating these datasets requires an enormous effort due to the need to analyze pure standard materials for each compound. Yet, the accuracy of these spectral annotation approaches has improved over the years[15,16]. This accuracy, understood as the ability to rank the correct identity of an unknown MS/MS within the top three candidates, has been shown to vary from 50% to 80% with state-of-the-art computational methods[12,17]. This increase of success in computational MS/MS spectra annotation can partly be attributed to the growing number of publicly available spectral data in libraries.

Another specific application of ML in small molecule discovery and analysis has been the prediction of chromatographic retention time (RT)[18]. Multiple ML-based RT prediction models, usually known as as quantitative structure–retention relationship (QSRR) models, have been reported over the last decade. These models typically aim at finding a relation between theoretical molecular descriptors or fingerprints with experimental RT. Similarly to in silico spectra prediction, the lack of experimental RT datasets has hampered the design of ML methods for large-scale RT prediction. Whereas community efforts now enable the access to thousands of MS/MS spectra[19], RT predictions have been largely based on small datasets, often not publicly available, ranging from a few hundreds[20–27] to less than 2200[28–32] molecules. Large datasets containing peptide RT data exist[33], but the only large-scale resource covering up to 114,000 unique small molecules is the commercial NIST retention index (relative RT) library. This library was designed for gas chromatography–mass spectrometry (GC–MS) and it is inapplicable to LC–MS.

Combined with mass spectrometry data, RT can be used as an additional and orthogonal evidence layer for small molecule putative identification, i.e., annotation[15,28]. In LC–MS, a computational model capable of predicting the RT for any molecular structure would enable filtering candidates with similar spectra but different RT. However, even in the ideal case of a perfect prediction model, these models only predict the RT for a specific chromatographic method (CM). Each laboratory usually uses a particular CM based on their specific needs, e.g., customizing solvent composition, gradient profile or flow rate, among others conditions. Although predicting RTs for a single CM lacks, in principle, scalability to other CMs from other laboratories, it has been shown that RT from one CM can be projected to other CMs given the general conservative compound elution order in reverse-phase (RP) columns[34–36]. This projection aims at making the RT of two CMs comparable. This implies that a prediction model for a particular method can be applied to other RP CMs as long as the compound elution order is well conserved between methods, thus making a specific prediction model scalable to other CMs.

Here we introduce the METLIN small molecule retention time (SMRT) dataset, a large-scale dataset consisting of experimentally acquired chromatographic RT covering 80,038 small molecules from the METLIN library[37] analyzed by RP liquid chromatography. To demonstrate the capability of the METLIN's SMRT dataset, we trained a deep-learning model for RT prediction. We analyzed the model's predictive ability along with the ability to project predicted RT onto other CMs. Further, we focused on the ability to rank and filter putative candidates in real metabolomics experiments based on accurate mass search and predicted RT.

## Results

**The METLIN SMRT dataset.** RP chromatography with high-performance liquid chromatography–mass spectrometry (HPLC–MS) was used to acquire RT data for a total of 80,038 small molecules (Fig. 1a) (see Methods for details). Pure standard materials for the 80,038 molecules were assembled by the California Institute for Biomedical Research and consisted of small molecules including metabolites, natural products and drug-like small molecules. These molecules can also be found in the METLIN library[37]. We used ClassyFire[38] to obtain a chemical taxonomy of molecules in the SMRT dataset. Most of molecules were classified into seven ClassyFire's groups (superclass level) including organoheterocyclic compounds (63.9%), benzenoids (24.7%), organic acids and derivatives (6.6%), organic nitrogen compounds (1.65%), organic oxygen compounds (1.18%), organosulfur compounds (0.66%) and other compounds (1.25%) such as lipids, lignans, nucleosides, nucleotides, phenylpropanoids and polyketides (Supplementary Fig. 1). The METLIN's SMRT dataset includes the RT in seconds, the PubChem numbers, the molfile containing the structures (SDF format), and molecular descriptors and extended connectivity fingerprints (ECFP) calculated with Dragon 7 (Kode Chemoinformatics, Pisa, Italy). See Data Availability section for information on dataset deposition.

**Application of deep learning for RT prediction.** We deployed a deep-learning regression model (DLM) using Keras for R[39] (see Methods for details on the DLM construction and parameters). From all the molecules in SMRT dataset, 75% of them were randomly selected and used as training set whereas the remaining 25% was used as validation set. ECFP[40,41] together with their respective RT were used as input data for the DLM (Fig. 1a).

Before choosing and configuring this DLM, we explored alternative input data and ML methods. We tested the performance of a DLM using molecular descriptors instead of fingerprints as input data. We used feature selection techniques to select relevant molecular descriptors and data normalization techniques to account for generalization bias[42]. We observed that fingerprints outperformed molecular descriptors. We also assessed other non-deep ML methods such as random forest regression using fingerprints. The random forest regression yielded a lower accuracy than the DLM, with mean and median error of 66 and 42 s, respectively (Supplementary Fig. 2). Finally, we explored the potential of 3D fingerprints[43] as an alternative to 2D fingerprints, which did not improve the prediction performance. We hypothesize that the more selective nature of the 3D fingerprints (they can distinguish between isomeric molecules) decreased the ability of the DLM to learn from similar molecules. Also, the influence of the molecular conformer used (i.e.,

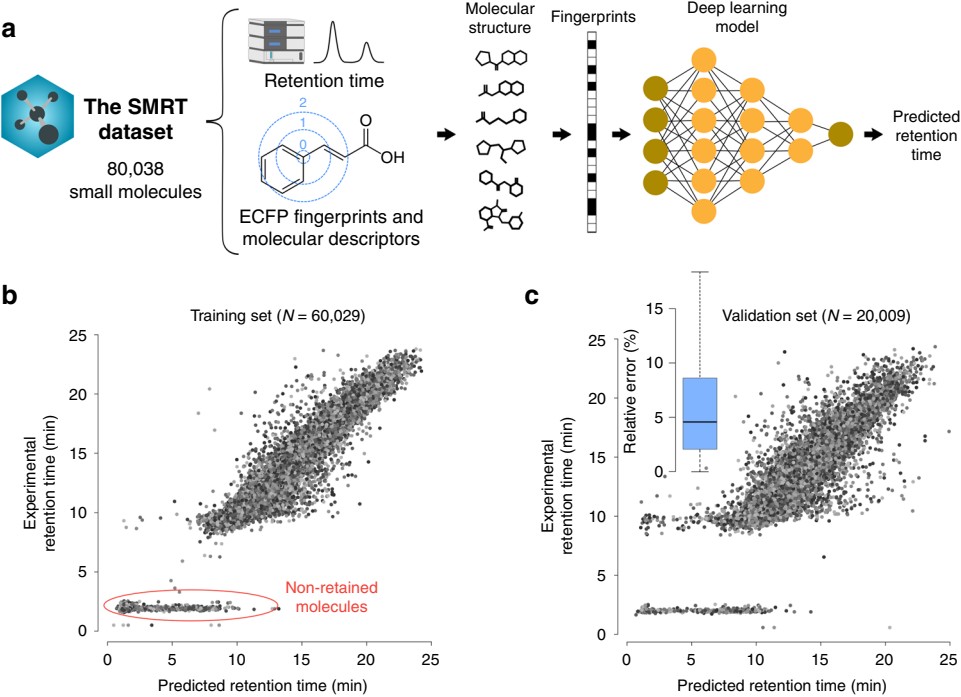

**Fig. 1 RT prediction results. a** Composition of the SMRT dataset and structure of the deep-learning model. **b** Predicted vs experimental RT for the training set and **c** validation set. Non-retained molecules are indicated (tentatively) in the training set plot. The relative prediction error box plot for the validation set is also shown. The box plot represents median value and interquartile range (25–75% percentiles) excluding outliers.

conformational isomer of each molecule) adds an additional complexity layer to the prediction process.

Prediction results of the proposed DLM for the training and validation set are shown in Fig. 1b, c. Figure 1c shows the overall validation set error, with mean and median relative errors of 47.8% and 4.6% and the mean and median absolute errors of 57 s and 35 s. To assess the magnitude of the error, we can compare this error with the mean chromatographic peak width of acquired molecules, which ranged between 20 and 30 s (full-width at half-maximum). This implies that a median error of 35 s can be considered low. The large difference between the mean and the median error is due to the presence of non-retained molecules in the model. Some compounds will not retain in the column and will elute before the start of the gradient, typically within the first minute. These non-retained molecules are not considered for data analysis in typical metabolomics experiments. The significant experimental RT gap between retained and non-retained molecules (Fig. 1b) induces a substantial error increase when the DLM fails at accurately predicting the RT for these non-retained molecules. If non-retained molecules are not considered, the gap between the mean and median error becomes significantly smaller, with mean and median relative errors of 6.8% and 4.5%. For the rest of the paper, non-retained molecules are not considered.

Literature describing RT prediction models usually reports prediction performance errors via $R^2$ values. These $R^2$ values depend largely on the size of the dataset used (number of molecules) and make performance comparison between studies difficult. Among studies reporting results in directly comparable values such as relative or absolute errors, Wolfer et al.[25] reported average relative and absolute errors of 13% and 23 s, respectively, in RP ultra-high performance liquid chromatography (UHPLC). In UHPLC, chromatograms are typically shorter and smaller absolute errors than in HPLC are expected. Falchi et al.[29] reported similar performance to our method with a median relative and absolute errors of 5% and 12 s in UHPLC, using a training set of 968 molecules.

First, we calculated the RT differences among similar molecules and evaluated the DLM's ability to accurately predict these differences. We used the Tanimoto similarity coefficient[44] to measure the similarity among molecules. The Tanimoto similarity coefficient measures how similar the 2D structures of two molecules are and it ranges from 0 (no similarity) to 100% (identical molecules). To assess the DLM's ability to accurately predict the RT differences among similar molecules, we implemented a naïve RT prediction approach by assuming that similar molecules will have similar RT. In that sense, a molecule's RT could be approximated with the median RT of other similar molecules. We implemented this naïve median RT approach using a $k$-nearest neighbors ($k$-NN) regression: for each molecule in the validation set, the $k$-NN searched the training set for the $k$ most similar molecules (we set $k$ to 3). Then, we determined the error between the molecule's experimental RT and the median RT of all the $k$ most similar molecules in the training set. Next, for all the molecules for which the $k$-median error was computed, we calculated the error between the experimental RT and the DLM's predicted RT. To further consider the effect of molecules' similarity, we compared the naïve and DLM's prediction error in five groups, each group considering those molecules in the validation set that had at least one similar molecule in the training set above a specific similarity threshold. The thresholds used were 95, 90, 80, 70 and 50%. The lack of molecules in the set with higher similarity scores (e.g., 99% or 98%) prevented the use of higher thresholds. Of note, nearly all molecules (99%) in the validation set had at least one molecule in the validation set with a 50% of similarity or more. We used the naïve $k$-NN approach as a control case to show whether or not an ML approach (DLM) can outperform a heuristic approach based on a general assumption ($k$-NN). Figure 2 shows the relative RT error for both the naïve approach (N) and the DLM predicted values (P). No statistically significant differences of exactitude (mean error) were observed for the 95% threshold (paired Wilcoxon rank test) although statistically significant differences in accuracy (standard

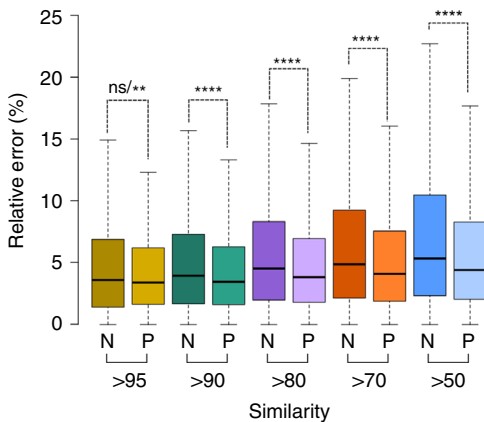

**Fig. 2 Naïve *k*-NN approach and DLM approach comparison.** Box plots for the naïve *k*-NN (N) and DLM (P) relative RT prediction error across similarity thresholds are shown. *P* values from paired Wilcoxon rank tests are summarized with asterisks ($n_{95} = 648$, $n_{90} = 6018$, $n_{80} = 14028$, $n_{70} = 16974$, $n_{50} = 19817$) except for the 95% threshold, for which ns (not significant) corresponds to the Wilcoxon rank test whereas the asterisks correspond to the Ansari–Bradley *P* value. Box plots represent median value and interquartile range (25–75% percentiles) excluding outliers.

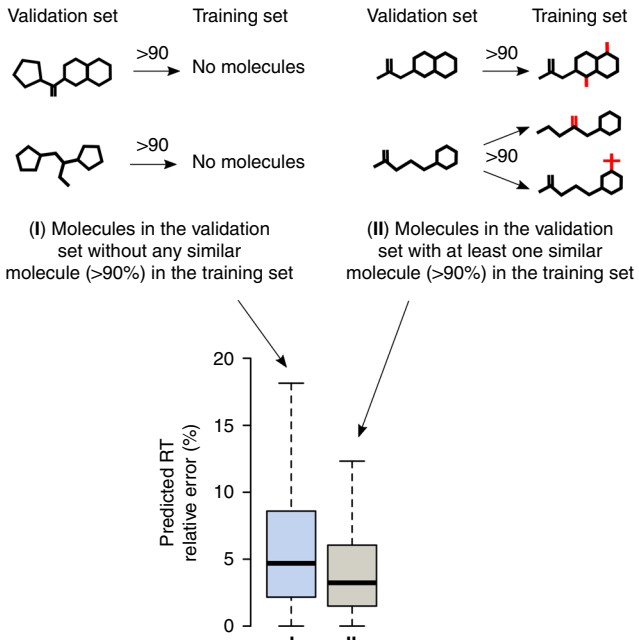

**Fig. 3 Prediction performance dependency on the training set.** The prediction error depends on the number and similarity level of molecules in the training set. To assess this dependency, the predicted RT error of molecules in the validation set having at least one similar molecule in the training set (more than 90% of similarity) was compared to molecules in the validation set with no similar molecule in the training set. Box plots represent median value and interquartile range (25–75% percentiles) excluding outliers.

deviation) were observed (Ansari–Bradley paired test, $P < 0.01$, $n = 648$). For the remaining thresholds, statistically significant differences in mean error were observed between the median approach and the DLM (paired Wilcoxon rank test, $P < 0.0001$, $n_{90} = 6018$, $n_{80} = 14028$, $n_{70} = 16974$, $n_{50} = 19817$).

The DLM was able to predict RT for similar molecules more accurately than the naïve *k*-NN approach ($IQR_P = 4.6\%$ compared to $IQR_N = 5.5\%$ for the 95% threshold and $IQR_P = 6.3\%$ compared to $IQR_N = 8.1\%$ for the 50% threshold). Yet, the (statistically significant) differences in mean error (exactitude) were relatively small ($median_P = 3.5\%$ compared to $median_N = 3.9\%$ for the 90% threshold and $median_P = 4.4\%$ compared to $median_N = 5.3\%$ for the 50% threshold). The fact that a naïve approach that considered only molecular similarity was capable of achieving a similar exactitude to the DLM suggests a strong influence of the structural similarity in the RT prediction. Previous studies have already observed that RT prediction performance improved when the training set included structurally similar molecules to those in the validation set[45,46]. Still, for highly similar molecules (similarity of 95% or above), results showed that there are no statistically significant differences between the predicted and naïve approach mean error (Fig. 2). This implies that it is more challenging for the DLM to accurately predict RT differences among highly similar molecules. We hypothesize that a training set with a greater number of groups of highly similar molecules is needed to allow the DLM to accurately predict RT differences among highly similar molecules.

To collect more evidence to support the hypothesis that a larger number of similar molecules in the training set increases the model's prediction performance, we tested whether the number of similar molecules in the training set had an impact on the RT prediction error (see Methods for details). We observed that when a molecule in the validation set had at least one similar molecule in the training set (similarity of 90% or above), the prediction error for that molecule was statistically significant smaller than those molecules without a similar molecule in the training set ($P < 0.0001$, $n = 5607$, Fig. 3). The same effect was observed for the remaining of similarity thresholds (80%, 70% and 50%, $P < 0.0001$, $n_{80} = 5425$, $n_{70} = 2718$, $n_{50} = 177$) but it was not observed for a 95% similarity. We hypothesize that two or more highly similar molecules (>95% similarity) are necessary

in the training set to observe a decrease in error in the case of 95% similarity or above. The lack of molecules with two or more highly similar molecules in the training set prevented determining the minimum number of molecules needed to observe statistically significant changes. This demonstrates that the ML-based RT prediction performance depends on the number and similarity level of molecules in the training set to those in the validation or a real case set.

**Projection of predicted RT in other CMs**. We studied the scalability of the prediction model by assessing how well predicted RT in our CM was able to predict RT in other CMs via predicted–experimental projections. We used the PredRet database[36] as a reference, which comprises small molecule experimental RTs for different CMs reported in the literature and generated in independent laboratories.

First, we analyzed the DLM's ability to accurately predict RT for biologically relevant metabolites. We predicted the RT for the 6823 molecules with KEGG number in The Human Metabolome Database (HMDB)[47]. Next, we selected the four RP CMs in the PredRet database that contained the largest number of molecules with KEGG number—we will refer to them as experimental chromatographic methods (ECMs). These methods are known in the PredRet database as FEM_long (ECM 1), LIFE_old (ECM 2), RIKEN (ECM 3) and FEM_orbitrap_plasma (ECM 4). ECMs 2 and 3 are short CMs (<5 min) whereas ECMs 1 and 4 are long CMs (25 min or more). To make the predicted and each ECM RT comparable, we projected predicted RT onto experimental RT[35] using predicted–experimental projections (see Methods for details). This projection yielded a non-linear function that allowed the RT interchange between the two CMs, i.e., by knowing the RT of a metabolite in one CM, we can predict the

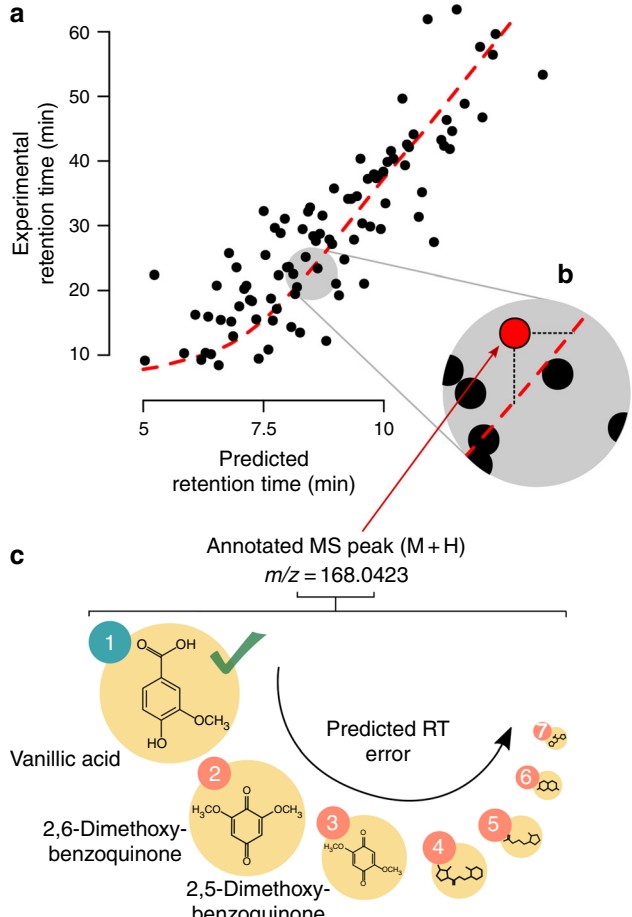

**Fig. 4 RT projection scheme. a** The RT of identified compounds eluting in two chromatographic methods (CMs) can be used to build a projection function (red dashed line). We determined this projection function by comparing predicted RT values (in the CM used to acquire the METLIN's SMRT dataset) to experimental RT values in a different CM (predicted–experimental projection). **b** This function enables projecting all the predicted RTs from their specific CM onto other experimental CMs. **c** By performing an accurate mass search, we can determine all possible known structures that could be attributed to any annotated protonated (M +H) or deprotonated (M−H) ion $m/z$. Predicted RT allows ranking this putative candidates by prediction error.

expected metabolite's RT in the other CM, and vice versa (Fig. 4a). To accurately project from one method onto another, a specific number of molecules need to be identified in both methods[36]. In our projection, we considered 50 randomly selected molecules in the ECM to simulate a scenario in which a fraction of the metabolite identities is known and confirmed. In the next step, we considered the protonated/deprotonated ion for each compound in the ECM without taking molecules' identities into account. This is done to simulate a common situation in untargeted metabolomics in which the protonated/deprotonated ion peak is observed (annotated) but the molecule's identity is not known. Typically, protonated/deprotonated ions of underlying metabolites in samples can be retrieved through a computational annotation process[15]. We used the mass determined from each metabolite's protonated/deprotonated ion to perform an accurate mass search (10 ppm mass error) against the HMDB to retrieve all the possible molecules with KEGG number that could be attributed to that protonated/deprotonated ion. We then compared the predicted RT for those molecules with their experimental RT (Fig. 4b). Here we aim at studying how

predicted RT can be used to rank potential identities and filter as many false-positive identities as possible (Fig. 4c). Of note, ECMs 1 to 4 had 4, 2, 3 and 4 molecules, respectively, that were included in the DLM's training set.

Results from these projections are shown in Fig. 5. The figure shows the four CMs (ECM 1–4) with five panels each. Panels a, f, k, p (Fig. 5) show the projection between the experimental and predicted RT. Relative errors between experimental and predicted RT for these ECM are shown in panels b, g, l, q (Fig. 5). These errors vary depending on the length of each ECM; the median error spans from 8 and 10% in short ECM (ECMs 3 and 2, respectively) to 14 and 17% in the remaining ECMs (ECMs 4 and 1, respectively). In seconds, this median absolute error is 5.7, 9.7, 89.4 and 210 s for ECM 3, 2, 4 and 1, respectively.

Experimental–experimental projections, i.e., those that exclusively project experimental RT from one CM onto another, have reported average relative errors of 2.6%[34,36]. Greater errors are expected in predicted–experimental projections in comparison with experimental–experimental projections mainly due to the inherent prediction error but also due to the projection error itself, which propagates the predicted RT error. In another study using the PredRet database to train and compare different ML-based prediction methods[27], models specifically designed for each ECM yielded median absolute errors ranging from 150 to 250 s for ECM 1 and from 7 to 14 s for ECM 3. Our predicted–experimental projections yielded median absolute errors of 210 and 5.7 s for ECM 1 and 3, respectively. In that sense, our predicted–experimental projections showed excellent performance compared to ML models generated from small datasets from individual CMs.

Next, we focused on the predicted RT ability to filter putative identities of annotated protonated/deprotonated ions in cases where there are multiple matches. After projecting predicted metabolite RTs onto each ECM, we determined an RT error threshold to filter metabolite identities, i.e., those metabolites with experimental–predicted RT difference under the error threshold will be considered correct identities and those above the threshold will be considered incorrect. To determine the best error threshold, we used a receiver operating characteristic (ROC) curve. We calculated the ROC curve by discretizing the range of relative RT error from 0 to 100% in intervals of 2.5%. For each error interval, we calculated the true positives (TP), true negatives (TN), false positives (FP) and false negatives (FN). Then, we determined the accuracy or true-positive rate (TPR) and the specificity or false-positive rate (FPR) (see Methods for details on the calculation of TP, TN, FP, FN, TPR and FPR). Panels d, i, n, s (Fig. 5) show the ROC curve for each ECM, with areas under the curve (AUC) ranging from 0.65 to 0.69 for all ECM.

The best filtering error thresholds determined by the ROC curves were 27.5%, 12.5%, 10% and 27.5% for ECM 1 to 4, respectively. This optimal threshold is the one that maximized the TPR while minimizing the FPR and, in effect, enabled filtering out as many FP by compromising as few TP as possible. The quantitative filtering performance, which can be assessed by comparing the total number of putative molecule candidates before and after filtering, is shown in panels c, h, m, r (Fig. 5). This filtering also incorrectly categorized a fraction of correct molecule identities (TP) as FP since they fall above the filtering threshold. Specifically, 31%, 41%, 36% and 21% of correct molecule identities for ECM 1 to 4, respectively, were considered as FP and filtered out. Altogether, ROC curves and their respective AUC values together with quantitative filtering results (Fig. 5c, h, m, r) show a moderate filtering capacity when using predicted–experimental projections.

Finally, we assessed the DLM's ability to rank putative identities, i.e., the ability to rank candidates from the most to

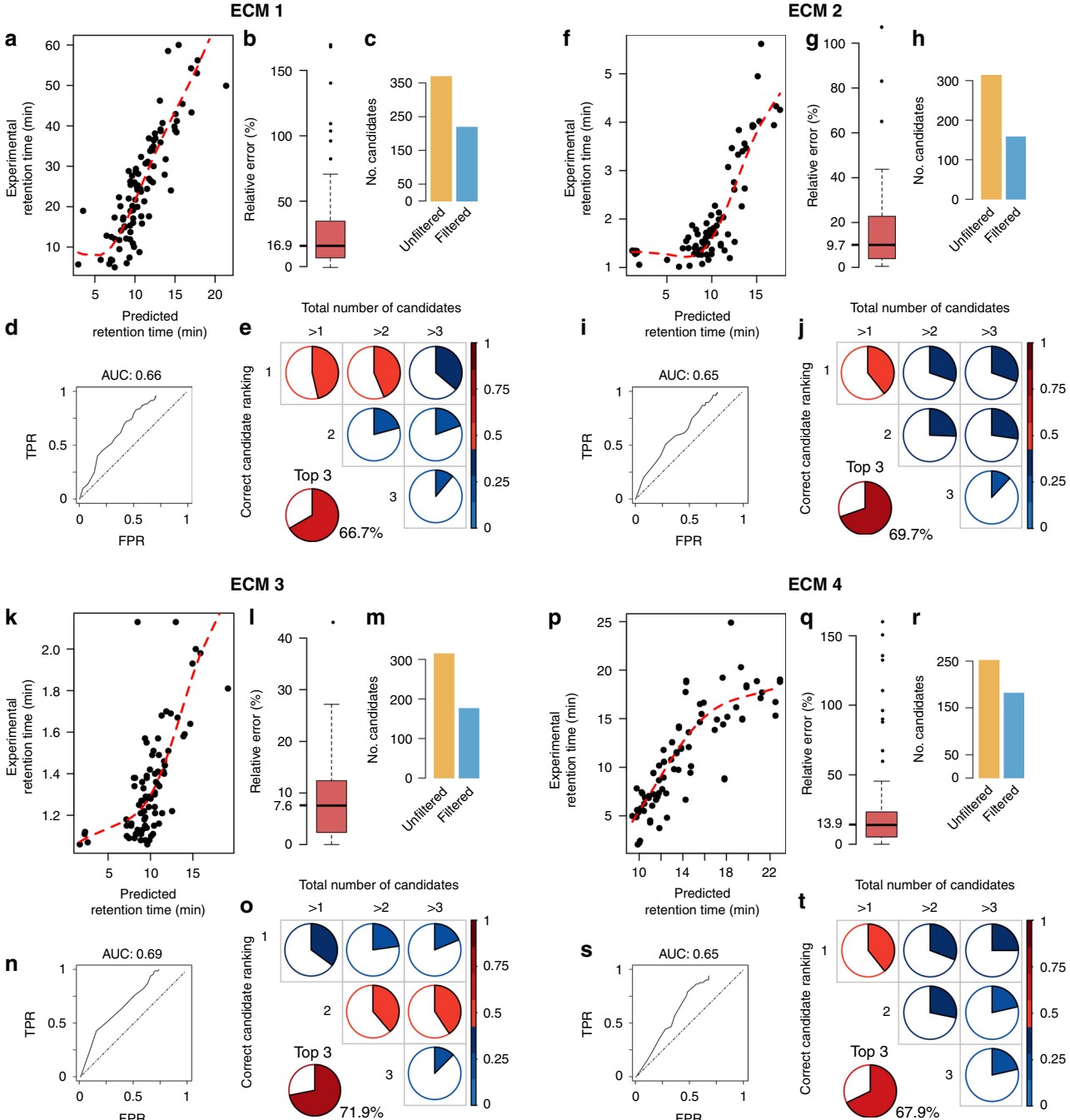

**Fig. 5 Predicted–experimental projections results.** Four experimental chromatographic methods (ECM 1 to 4) with five panels each. In the PredRet database, ECMs 1–4 are known as FEM_long, LIFE_old, RIKEN and FEM_orbitrap_plasma, respectively. Panels **a**, **f**, **k**, **p** show the projection between the experimental and predicted RT. Panels **b**, **g**, **l**, **q** show relative errors between the experimental and predicted RT. Box plots represent median value and their error bars represent the interquartile range (25–75% percentiles). Panels **c**, **h**, **m**, **r** show the number of putative candidates before and after filtering by RT error threshold. Panels **d**, **i**, **n**, **s** show the ROC curves with their respective AUC values. Panels **e**, **j**, **o**, **t** show the candidate ranking. Ranking panels show the percentage of molecules in which the correct identity is exactly the first, second or third top candidate (y-axis, correct candidate ranking) and there are more than 1, 2 or 3 putative candidates in total (x-axis, total number of candidates). The top-three chart shows the percentage of cases in which the correct identity was ranked among the top three candidates of the total of cases with more than three candidates.

the least likely identity according to their RT error, with a special focus on the ability to rank the correct identity among the top three first candidates (Fig. 4c). Panels e, j, o, t (Fig. 5) show the ranking results. These panels show the percentage of molecules in which the correct identity was ranked as the first, second or third candidate (y-axis) and whether there were more than 1, 2 or 3 putative candidates in total (x-axis). Percentages were calculated

out of the total number of cases with more than 1, 2 or 3 candidates in each case. The top three chart shows the percentage of cases in which the correct identity was ranked among the top three candidates of the total of cases with more than three candidates. Ranking results showed that predicted RT enabled ranking the correct identity among the top three candidates in ~70% of the cases for all methods. This demonstrates an excellent

ranking capacity at the same level as current MS/MS annotation algorithms[12,17]. An example of a successful ranking is shown in Fig. 4c, where an observed experimental protonated/deprotonated ion could be attributed to multiple unique molecular identities, some of which were structurally similar. Based on RT error, the correct molecule's identity, vanillic acid, was ranked first.

Further, we performed predicted–experimental projections for all remaining ECMs in the PredRet database that contained at least 20 molecules with KEGG number (Supplementary Figs. 3–15). These results are in good alignment with those observed for ECM 1 to 4, with AUC for the ROC curve all above 0.6 and with ranking performance above 50%. The differences between the CMs being projected and the low number of identified molecules (experimental RT data points) in the ECMs negatively impacted the performance for some methods such as MTBLS38 or UniToyama_Atlantis (Supplementary Figs. 10 and 15).

When performing predicted–experimental projections, we searched for putative identities within a relatively small number of molecules (a total of 6823 molecules with KEGG number). When this search space was increased, covering the 18,457 molecules in HMDB with PubChem number, the overall ranking and filtering performance of the predicted–experimental projection strategy decreased as expected. Specifically, AUC values for the ROC curves were comprised between 0.57 and 0.58 for ECM 1 to 4 and top three ranking performance was comprised between 41% and 45% for all cases (Supplementary Fig. 16). This is due to the fact that a specific protonated/deprotonated ion can match to a larger number of molecules, making the ranking of the correct identity more challenging than when fewer candidates are considered. The use of MS/MS data (in combination with other orthogonal properties like RT) is required for a confident identification[48]. However, our predicted–experimental RT projections—based only on accurate mass matching and predicted RT—showed a significant ranking capacity without the use of MS/MS data. We anticipate an enhanced ranking performance by combining RT prediction with MS/MS libraries[13] or with advanced annotation approaches like computational MS/MS spectral annotation[12] or MS[1] in-source annotation[49].

Overall, results demonstrated that a DLM generated with the METLIN's SMRT dataset accurately predicts RT for a large number of molecules. We demonstrated that this RT prediction model is scalable and applicable for the annotation of small molecules in other CMs from independent laboratories. RT projections require knowing the identity and RT of a specific number of molecules in both the CMs being compared. For ECM 1 to 4, we randomly selected a subset of molecules (50) to calculate the projection function. Results varied each time that the projection was calculated due to this random selection mainly due to aberrant RT predictions. By computing several iterations, we observed significant variations mainly in the value of the filtering error threshold, which reinforces our conclusion that predicted–experimental RT filtering is not a robust approach. Overall prediction error and ranking results yielded similar values across iterations. Therefore, these projections need supervision to ensure that they are not biased. Alternative projection methods based on taking into account the CM gradient and flow rate differences[29,50] have been shown to be more robust because the molecules' RT is not considered in the determination of the projection function. Yet, these methods require a comprehensive characterization of each CM.

The METLIN's SMRT dataset provides RT data for RP chromatography. RP is one of the two most widely used chromatography methods along with hydrophilic interaction (HILIC). HILIC chromatography provides complementary separation to RP, enabling the separation and measurement of molecules that might not be retained in RP. As opposed to RP, the elution order among different HILIC CMs is not well-conserved. Since a similar elution order is needed for accurate RT projections between CMs, the use of HILIC compromises both experimental–experimental and predicted–experimental projections when two different HILIC-based CMs are being compared.

In conclusion, the METLIN's SMRT dataset enabled accurate RT prediction through a deep-learning approach with median relative and absolute errors of 4.6% and 35 s (HPLC), respectively. We showed that predicted RT generated with a training set from a particular CM can be projected onto other CMs. Results demonstrated that predicted–experimental projection has a significant capability to rank the correct identity among the top three putative candidates with the same formula mass. Specifically, predicted–experimental projection ranked the correct identity among the top three candidates in up to 70% of cases. Results also demonstrated that ML-based prediction performance is limited by the number and similarity level of molecules in the training set to those in the validation or a real case set. A greater prediction error is expected for those molecules lacking structurally similar molecules in the training set. Yet, the SMRT dataset provides sufficient data to enable the design of alternative first principles or ab initio strategies (e.g., based on quantum chemistry) to overcome natural ML limitations.

Collectively, the METLIN's SMRT dataset provides the community with a large-scale dataset for RT prediction, enabling a more confident metabolite annotation with broad applications in pathway enrichment or natural language processing (NLP) platforms—which digest scientific literature to rapidly interpret complex datasets within biological contexts. The use of ML, including NLP for scientific literature interpretation, represents a future direction for metabolomics research. We anticipate that, in the same way that the availability of MS/MS spectral datasets has improved the performance of in silico MS/MS prediction algorithms, the METLIN's SMRT dataset will contribute to enhancing the accuracy of RT prediction models.

## Methods

**Dataset assembly and RT acquisition by LC–MS.** The pure standard materials for the 80,038 molecules were analyzed in batches composed of mixtures of ~100 molecules with different molecular weight. Pure standard materials were analyzed by HPLC on an Agilent 1100/1200 series liquid chromatography (LC) system coupled to a quadrupole-time of flight (Q-TOF) mass spectrometer G6538A (Agilent Technologies, Santa Clara, CA) using a Zorbax Extend-C18 reverse-phase column (2.1 × 50 mm, 1.8 $\mu$m, Agilent Technologies, Santa Clara, CA). The gradient consisted of 5% B for 3 min, 50% B over 2 min, 85% B over 15 min and held at 85% B for 3 min, with a flow rate of 100 $\mu$L/min. All analyses were performed in positive and negative ionization mode. The composition for the mobile phases A and B consisted of water + 0.1% formic acid and acetonitrile + 0.1% formic acid, respectively. Dead and dwell volume were 40 $\mu$L and 900 $\mu$L. Resulting molecules' protonated/deprotonated or other adducts peaks were computationally extracted via a peak-picking approach. All peaks were manually inspected prior to RT integration to the database. RT data were acquired over the course of 3 months and certain RT variability is expected. We used a subset of 198 molecules to determine the RT variability. Throughout the analysis of molecules' batches, we randomly selected and analyzed one of the reference molecules each time that a new batch was analyzed. Each molecule was analyzed at least twice with a difference of at least 30 days and we computed the RT variability of the same molecule. We observed a mean and median RT variability of 36 and 18 s, respectively.

**Deep-learning model construction and parameters.** A deep-learning model using Keras for R[39] was deployed following the typical scalar regression configuration[42] and consisted of 4 fully connected hidden layers with 1000, 500, 200 and 100 units, respectively, activated by a non-linearity function (relu), connected to an output layer consisting of one unit with no activation. Regularization was performed by a L2 regularizer and the regulation parameter was set to 0.0001. An adam optimizer was used with a learning rate of 0.01 and mean squared error was used as loss function. A total of 20 epochs were used to train the model with a batch size of 35. Training and validation set molecules were randomly selected. Results and statistical significance remained consistent across multiple iterations

with randomly selected training and validation sets. The sample size ($n$) denoted for each case in this paper varied depending on the training/validation set used. When a statistically significant difference was observed only for certain training/validation sets, we considered this difference as not statistically significant.

**Impact of the number of similar molecules on RT prediction error**. To test whether the number of similar molecules in the training set had an impact on the RT prediction error we performed the following procedure: we compared the RT error for those molecules in the validation set having at least one similar molecule in the training set above a specific similarity threshold to molecules with no similar counterpart (Fig. 3). This threshold varied and is described in the Results section. Since two groups with different sizes ($n$) were being compared (e.g., a group from molecules with and without similar counterpart in the training set), we tested for statistical significance using a subset of observations. For each case, the size of the subset corresponded to the smallest of the two group' sizes (sizes are reported in the Results section). To account for false null hypothesis rejection when using a sample subset, we repeated a Wilcoxon rank test 1000 times, each time selecting a different random subset of observations from the group with the largest size. Resulting $P$ values where further corrected using the Bonferroni method. The final statistical significance level (n.s., $P < 0.05$, $P < 0.01$, $P < 0.001$, $P < 0.0001$) was selected as the smallest significance level having at least 95% of the corrected $P$ values under the specific significance level.

**Predicted–experimental RT projection**. Predicted RT was projected onto experimental CMs via a robust polynomial regression (least squares regression of a polynomial function), using the function *poly* from the R package *stats*. The polynomial function (four polynomials) was adjusted by minimizing the error between the experimental RT of the molecules in the specific CM and the DLM's predicted RT of the same molecules. This yielded a projection function that allowed adjusting all the predicted RTs to make them comparable to the given CM. Only a random subset of molecules (50 molecules) from all the molecules in each CM were used for determining the projection function.

**ROC curve-associated parameters**. For each threshold used to determine the ROC curve, we calculated the number of true positives (TP), true negatives (TN), false positives (FP) and false negatives (FN). TP was the number of cases where correct identities' RT error was under the error threshold, TN was the number of cases where false identities' RT error was above the error threshold, FP was the number of cases where false identities' RT error was under the error threshold and FN was the number of cases where false identities' RT error was above the error threshold. The accuracy or true-positive rate (TPR) was calculated as FP/(FP+TN) and the specificity or false-positive rate (FPR) was calculated as TP/(TP+FN).

## Data availability

The METLIN's SMRT dataset is available in the Figshare repository: https://doi.org/10.6084/m9.figshare.8038913.

## Code availability

The R scripts to reproduce the results are available in the Figshare repository: https://doi.org/10.6084/m9.figshare.8038913.

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

## Acknowledgements

This research was partially funded by National Institutes of Health grants R35 GM130385, P30 MH062261, P01 DA026146 and U01 CA235493; and by Ecosystems and Networks Integrated with Genes and Molecular Assemblies (ENIGMA), a Scientific Focus Area Program at Lawrence Berkeley National Laboratory for the U.S. Department of Energy, Office of Science, Office of Biological and Environmental Research, under contract number DE-AC02-05CH11231. This research benefited from the use of credits from the National Institutes of Health (NIH) Cloud Credits Model Pilot, a component of the NIH Big Data to Knowledge (BD2K) program.

## Author contributions

G.S. led and supervised the project; X.D.-A. conceived and designed the study, interpreted the data and wrote the manuscript; E.C. assembled and provided the small molecules standard materials; W.U. and E.B. analyzed the pure materials via LC–MS; A.E.A. and H.P.B. computationally processed raw data to extract retention time data; C.G., E.B. and J.R.M.-B. contributed to data interpretation and provided revisions to the manuscript. All authors approved the final version of the article.

## Competing interests

The authors declare no competing interests.
