## [Peer Review File · Nature Communications]

Reviewers' comments:

Reviewer #1 (Remarks to the Author):

The article entitled « The SMRT dataset for retention time prediction of small molecules » represent a huge amount of work corresponding to the analysis of more than 80,000 compounds. Beyond providing such a database (DB), the authors evaluate the ability of such a data to be used to predict RT using deep machine learning. Using such strategy seems to outperformed any previously generic models provided up to now (either in direct prediction or prediction coupled with LC-method transfer, or by its use for annotation improvement), even though basic concept of liquid chromatography were not taken into consideration.

To my point of view such DB is very important to provide to the community as a starting point for further development in RT prediction.

I have some major point that should be added to the manuscript:

- The authors need to provide how they check RT reliability over 80,000 LC-MS analysis? Did the authors injected mix of standards? Or did they inject every compounds alone? This point need to be clarified to convinced potential user about its reliability.
- Please provide dead and dwell volumes of the chromatographic apparatus used for the data set acquisition. This is important to be able to used fundamental chromatographic concept for further used of this data set in future development in RT prediction.
- It was not clear to me how RT prediction with other method using PredRet database was achieved. In particular, how method transfer was achieved? This is important because column geometry and gradient was different between the original method and other methods called CM1 to CM4 in the manuscript.

These point need to be answered prior to any publication. In addition, I have some comments that could improve manuscript and result quality:

- Did the authors try to predict the RT predictability for structures with RT bellow 5 min?
- Could the authors provide a set of easily purchasable compounds that could be used for a better transposition of RT prediction on any LC systems using a similar column?

Minor comments that need to be corrected:

- Please replace UPLC which is the Waters brand by UHPLC (ultra-high pressure liquid chromatography) which correspond to the technology itself.
- - please split Fig1 into two figures corresponding to Fig 1a-c and Fig1de. This would improve clarity of the data set used for every part of the figure. In fig 1a-c all dataset was used, which was not the case for the other part of the figure.

Reviewer #2 (Remarks to the Author):

This paper makes two major contributions to research community

(a) it puts forward a large database, SMRT, of Liquid-Chromatography retention times of small molecules accompanied by their molecular structure information,

(b) it presents an application of a deep learning model to predict retention times for new molecules from their structure. Currently, metabolite identification is a major bottle-neck in metabolomics, as the identify (the molecular structures) of ca. 90% of metabolites observed in a typical LC-MS run remain unknown. Compared to MS and MS/MS data, retention times remain an under used resource due to the unavailability of large enough datasets to train machine learning models, as well as due to the high variability of retention times across chromatographic systems.

With its ca. 80000 molecules, the SMRT database presents a step-change in the amount of publicly available data on retention behaviour of small molecules, compared to ca. 2000 molecules currently available. This kind of public data source is likely to lead to significant acceleration of science and lead to significantly improved metabolite identification rates.

As for the deep learning application, I consider it an excellent baseline against which different research groups can measure their performance. As such, the results are far from perfect for the deep learning model, but they serve the purpose of demonstrating the usefulness of the data source.

Below I have a few detailed comments and questions for improving the manuscript:

- The intro is rather skimpy on the references to papers that have previously tackled retention time or retention order prediction. I would recommend adding more references to previous methods.
- page 2, "Application of deep learning". How was the training/test split implemented. Was it completely random or was there some effort to make sure that the distributions of molecules were somehow similar in the train and test fold?
- page 4, left column, second para. I am not fully convinced with this argumentation. Your results show that the median approach is more accurate than deep learning at 95% Tanimoto similarity. Indeed, the median approach gets weaker if lower threshold is used but I think this may not be a meaningful comparison. Given a new molecule, I could easily take the median on k=3 nearest neighbors, and thus circumvent the need to have a fixed threshold. I wonder if this simple knn-regressor would be actually very competitive with the deep learning model.
- page 4, right column, "Application of ...": did you remove the 26 molecules from the dataset, I think that would be advisable.
- "the experimental CM" is ambiguous at first reading - it could mean your CM or the predret CMs.
- I think that instead of Fig 2 (which is of not that high information content) you should add a figure depicting the protocol by which you obtain the predicted retention times for the KEGG molecules.
- page 5, right column, second para. "predicted RT under the error threshold" is not an accurate expression, you mean the experimental-predicted RT difference under the error threshold.
- page 7, left column, third para: I am not sure what you want to say with the sentence "However, ranking ...". Your method is also using MS1 and RT...Do you mean *database lookup* with MS1 and RT is a weak method (I would agree with that).
- left column, second para. Again, I am not sure what you want to say here. That it is not feasible to use similar ML approach for HILIC? I am not sure I agree without evidence, there might even be bigger room for ML there since the relationships are more complex.

Reviewer #3 (Remarks to the Author):

As the title suggests the authors contribute a retention time (RT) of small molecules dataset. This dataset contains the experimental RT for more than 80,000 small molecules represented by several features. This is a significant increase over what is currently publicly available and should improve future efforts to exploit RT to differentiate small molecules with very similar structures. As such this contribution is highly valuable for small molecule research and justifies the publication of the manuscript.

However, I do have some major concerns that need to be addressed.

1. The manuscript does not provide enough detail on the creation of the dataset. It mentions "pure standards materials where assembled" but it is unclear what this means. Where were the 80,038 pure standards? If not, how were small molecules identified? Describing the procedure in more detail would be a valuable (if not critical) addition to the manuscript.
2. The authors demonstrate the value of the dataset by training a neural network with four hidden layers on the data that predicts the RT given a feature vector representation of a small molecule. From Machine Learning we know that "deep" neural networks can benefit significantly from big datasets where other "classical" learning algorithms cannot. It is however unclear if this is the case here. The few comparisons made with other research results show only slight improvements and, in some cases, even a performance decrease (CM1). I believe this should be discussed in more detail.
3. The authors present learning curve results showing statistically significant improvements for an increasing number of training molecules up to 20,000. The authors do not discuss hyperparameter optimization for the smaller training sets. Were these fixed for all training sets? Does that then not bias the conclusions?
4. Given the large number of small molecules in the dataset it is very likely that test sets (like CM1 and CM3) contain molecules that are also in the training set. If so this is a problem as we can expect overly optimistic results when evaluating generalization performance.

Minor concerns:

1. The authors use statistical tests to address the significance of the small differences they observe. With the sample sizes used in the tests even the smallest difference becomes significant. I think the authors should state more clearly that even though the tests show significant differences, these differences are indeed very small.
2. The authors state that R² values cannot be compared between datasets with different sizes. Why is that?

Point by point response to reviewers' comments of manuscript (NCOMMS-19-14751) entitled "The SMRT dataset for retention time prediction of small molecules".

We would like to thank the three referees for emphasizing the full significance of our study and for their constructive criticisms and comments that we feel have substantially improved the quality of our manuscript.

Reviewers' comments:

Reviewer #1 (Remarks to the Author):

The article entitled « The SMRT dataset for retention time prediction of small molecules » represent a huge amount of work corresponding to the analysis of more than 80,000 compounds. Beyond providing such a database (DB), the authors evaluate the ability of such a data to be used to predict RT using deep machine learning. Using such strategy seems to outperformed any previously generic models provided up to now (either in direct prediction or prediction coupled with LC-method transfer, or by its use for annotation improvement), even though basic concept of liquid chromatography were not taken into consideration.

To my point of view such DB is very important to provide to the community as a starting point for further development in RT prediction.

I have some major point that should be added to the manuscript:

- The authors need to provide how they check RT reliability over 80,000 LC-MS analysis? Did the authors injected mix of standards? Or did they inject every compounds alone? This point need to be clarified to convinced potential user about its reliability.

Answer: We thank the reviewer for raising this concern. We analyzed the standards in batches composed of mixtures of approximately 100 molecules with different molecular weight. Retention time data were acquired over the course of 3 months and certain RT variability is expected. We used a subset of 198 molecules to determine the RT variability. Throughout the analysis of the molecules' batches, we randomly selected and analyzed one of the reference molecules each time that a new batch was analyzed. Each molecule was analyzed at least twice with a difference of at least 30 days and we computed the RT variability of the same molecule. We observed a mean and median RT variability of 36 and 18 seconds, respectively. We clarified this in the Methods Section.

- Please provide dead and dwell volumes of the chromatographic apparatus used for the data set acquisition. This is important to be able to used fundamental chromatographic concept for further used of this data set in future development in RT prediction.

Answer: Dead and dwell volumes were experimentally calculated and are now included in the Methods section. Specifically, dead and dwell volumes were 40 uL and 900 uL.

- It was not clear to me how RT prediction with other method using PredRet database was achieved. In particular, how method transfer was achieved? This is important because column geometry and gradient was different between the original method and other methods called CM1 to CM4 in the manuscript.

Answer: Method transfer was carried by RT projection, which has been shown to be a reliable method to transpose RT between chromatographic methods with different columns or solvents (Abate-Pella, D. et al. doi: 10.1016/j.chroma.2015.07.108; Stanstrup et al doi: 10.1021/acs.analchem.5b02287). An overview on the use of this approach is included also in the introduction. Figure 4 in the manuscript also depicts this projection process. We agree with the reviewer that, in the Results section, the description was somewhat confusing and we have clarified it. We also added a more technical description in a 'Predicted-experimental retention time projection' subsection in the Methods section describing how RT projection was performed. In addition, and to facilitate reproducibility, the script with the RT projection method is included as supplementary materials (see Data availability section)

These point need to be answered prior to any publication. In addition, I have some comments that could improve manuscript and result quality:

- Did the authors try to predict the RT predictability for structures with RT bellow 5 min?

Answer: We thank the referee for this excellent suggestion. Since peaks from non-retained molecules are usually not considered for data analysis (e.g., statistical analysis, annotation, etc) we did not consider building a specific prediction model for these molecules. However, we thought that a more useful alternative would be creating a model that predicts whether a molecule will or will not be retained by the column using only the experimental data in the SMRT database (see **Attached Figure 1**). We did not include these results in the manuscript because validating these results with other experimental data is challenging. This is because it is difficult to determine what is a retained or non-retained molecule in the PredRet's experimental CMs, and this could bias the results.

Attached Figure 1. A density of the predicted RT of non-retained molecules (marine blue, left) and retained molecules (cyan blue, right). We used the experimental RT to classify molecules between retained (RT above 5 min) and non-retained (RT below 5 min). A naïve Bayesian classifier could be used to classify any given molecule between retained

and non-retained and, based on the densities of both groups, we observe that a majority of the molecules with a predicted RT below 5 min could be classified correctly as non-retained molecules, and vice versa.

- Could the authors provide a set of easily purchasable compounds that could be used for a better transposition of RT prediction on any LC systems using a similar column?

Answer: The referee makes a valid point to enhance the accuracy of RT projection between the SMRT chromatographic method (CM) and other CMs using a set of molecules as a reference. However, it is challenging to select a set of molecules that span all the chromatogram and that are easily available. Also, given previous studies together with our findings, we believe that RT projection using any observed molecules in any given CM (predicted-experimental projection) is an accurate and easy-to-perform alternative for RT transposition between different chromatographic methods. As an example, the method that the referee is proposing could not be applied for the validation of our study with the CMs in the PredRet database.

Minor comments that need to be corrected:

- Please replace UPLC which is the Waters brand by UHPLC (ultra-high pressure liquid chromatography) which correspond to the technology itself.

Answer: This has been corrected.

- - please split Fig1 into two figures corresponding to Fig 1a-c and Fig1de. This would improve clarity of the data set used for every part of the figure. In fig 1a-c all dataset was used, which was not the case for the other part of the figure.

Answer: The suggested changed has been made.

Reviewer #2 (Remarks to the Author):

This paper makes two major contributions to research community (a) it puts forward a large database, SMRT, of Liquid-Chromatography retention times of small molecules accompanied by their molecular structure information, (b) it presents an application of a deep learning model to predict retention times for new molecules from their structure. Currently, metabolite identification is a major bottle-neck in metabolomics, as the identify (the molecular structures) of ca. 90% of metabolites observed in a typical LC-MS run remain unknown. Compared to MS and MS/MS data, retention times remain an under used resource due to the unavailability of large enough datasets to train machine learning models, as well as due to the high variability of retention times accross chromatographics systems.

With its ca. 80000 molecules, the SMRT database presents a step-change in the amount of publicly available data on retention behaviour of small molecules, compared to ca. 2000 molecules currently available. This kind of public data

source is likely to lead to significant acceleration of science and lead to significantly improved metabolite identification rates.

As for the deep learning application, I consider it a excellent baseline against which different research groups can measure their performance. As such, the results are far from perfect for the deep learning model, but they serve the purpose of demonstrating the usefulness of the data source.

Below I have a few detailed comments and questions for improving the manuscript:

- The intro is rather skimpy on the references to papers that have previously tackled retention time or retention order prediction. I would recommend adding more references to previous methods.

Answer: The referee is right to point out that some references are missing. Our intention was to make the introduction concise by citing only some of the most representative papers reporting RT prediction models. In the revised version of the manuscript, we have added three more references (one reporting the retention order prediction) that could be of interest to readers.

- page 2, “Application of deep learning”. How was the training/test split implemented. Was it completely random or was there some effort to make sure that the distributions of molecules were somehow similar in the train and test fold?

Answer: The selection was completely random and we predicted the RT multiple times with a random training/validation set each time and reproduced the findings in this paper. We did not observe any difference for each selection (statistical significance of all the results remained the same). Consequently, we did not consider taking the distribution of the training set into account. This has now been clarified in the Methods section (Deep learning model construction and parameters).

- page 4, left column, second para. I am not fully convinced with this argumentation. Your results show that the median approach is more accurate than deep learning at 95% Tanimoto similarity. Indeed, the median approach gets weaker if lower threshold is used but I think this may not be a meaningful comparison. Given a new molecule, I could easily take the median on k=3 nearest neighbors, and thus circumvent the need to have a fixed threshold. I wonder if this simple knn-regressor would be actually very competitive with the deep learning model.

Answer: We thank the referee for this excellent observation. We also believe that the KNN approach is more meaningful and we have now changed the median approach for the KNN version (K=3). Qualitatively, results remain the same, i.e., the DLM outperforms the KNN approach even at 95% of similarity. Quantitatively, the KNN approach accuracy is

better compared to the median approach, but still far from the DLM accuracy (we refer to the accuracy as the SD of the error, and to the exactitude as the mean or median error). This can be explained because although the KNN approach will compute the median RT of the k most similar molecules, there is no guarantee that there are k very similar molecules in the training set to the those in the real case. Therefore, the KNN will yield inaccurate predictions when there are no similar molecules in the training set.

- page 4, right column, “Application of ...”: did you remove the 26 molecules from the dataset, I think that would be advisable.

Answer: We agree that this is an important point to consider. However, from these 26 molecules, only 13 were in the PredRet database and scattered across its different CM. Specifically, CM1 to 4 only have 4, 2, 3 and 4 molecules, respectively, that were included in the training set. We believe that removing this small number of molecules will not have an impact on the quantitative/qualitative results. We have clarified this in the manuscript.

- “the experimental CM” is ambiguous at first reading - it could mean your CM or the predret CMs.

Answer: We thank the referee for bringing up this point. We have now defined another acronym to refer to the experimental (PredRet) CM exclusively: experimental chromatographic method (ECM).

- I think that instead of Fig 2 (which is of not that high information content) you should add a figure depicting the protocol by which you obtain the predicted retention times for the KEGG molecules.

Answer: With all due respect to the reviewer, we feel that predicting the RT of all the KEGG molecules is a simple operation. Depicting this process in a figure would make its content redundant as it would inevitably imply the reuse of parts of other figures such as Fig 1a (top/right diagram, from molecule structure to fingerprints, to the use of the DLM). We agree that the former Figure 2 (currently Figure 4) does not show relevant information for experts in RT prediction, as they might already know how RT projection works, but we believe that the figure may help a broader audience to understand what RT projection is.

- page 5, right column, second para. “predicted RT under the error threshold” is not an accurate expression, you mean the experimental-predicted RT difference under the error threshold.

Answer: We thank the referee for pointing this out. This has been corrected.

- page 7, left column, third para: I am not sure what you want to say with the sentence “However, ranking ...”. Your method is also using MS1 and RT...Do you mean *database lookup* with MS1 and RT is a weak method (I would agree with that).

Answer: The referee is correct. We meant that the considering only MS¹ and RT is a weak annotation method. Despite that, we obtained good identification results using this “weak” method and by using predicted RT, so we expect better identification results when more sophisticated MS¹-based annotation methods or MS/MS information are combined with predicted RT information. We agree that this was somewhat confusing and we have revised the paragraph to make it clearer.

- left column, second para. Again, I am not sure what you want to say here. That it is not feasible to use similar ML approach for HILIC? I am not sure I agree without evidence, there might even bigger room for ML there since the relationships are more complex.

Answer: We thank the referee for pointing this out. What we are trying to convey is that if the same RT database had been acquired in HILIC, it would lack scalability to other CMs. This is because, compared to RP, the elution order between two different HILIC methods is not well-conserved. As discussed in the introduction, a necessary condition to allow projection between two CMs is that the elution order is well-conserved between the two methods being compared. We agree with the reviewer that this paragraph was confusing and we have now revised it to make it clearer.

Reviewer #3 (Remarks to the Author):

As the title suggests the authors contribute a retention time (RT) of small molecules dataset. This dataset contains the experimental RT for more than 80,000 small molecules represented by several features. This is a significant increase over what is currently publicly available and should improve future efforts to exploit RT to differentiate small molecules with very similar structures. As such this contribution is highly valuable for small molecule research and justifies the publication of the manuscript.

However, I do have some major concerns that need to be addressed.

1. The manuscript does not provide enough detail on the creation of the dataset. It mentions “pure standards materials where assembled” but it is unclear what this means. Where there 80.038 pure standards? If not, how were small molecules identified? Describing the procedure in more detail would be a valuable (if not critical) addition to the manuscript.

Answer: We thank the referee for the comment. We agree that this could be misleading and we have now clarified it making it explicitly clear that the 80.038 molecules analyzed were pure standard materials. We have described also in more detail (as requested by Referee 1) how the RT data was acquired (see Methods section, ‘Database assembly and retention time acquisition by liquid chromatography - mass spectrometry’).

2. The authors demonstrate the value of the dataset by training a neural network with four hidden layers on the data that predicts the RT given a feature vector

representation of a small molecule. From Machine Learning we know that “deep” neural networks can benefit significantly from big datasets where other “classical” learning algorithms cannot. It is however unclear if this is the case here. The few comparisons made with other research results show only slight improvements and, in some cases, even a performance decrease (CM1). I believe this should be discussed in more detail.

Answer: The referee is right and this is why in our study we minimized the importance of using a deep learning prediction approach and focused instead on the advantages of the prediction itself. We used a deep learning model because it was faster and more accurate than other alternatives that we tried. We did not achieve a comparable performance via a non-deep ML method like random forest regression model (see Supplementary Figure 2). However, it is difficult to demonstrate that there are no non-deep ML methods that could outperform our DL approach and this is why we do not make this claim in our paper. Despite that, we agree with the reviewer that this could be discussed and we have now included these observations in the manuscript.

3. The authors present learning curve results showing statically significant improvements for an increasing number of training molecules up till 20,000. The authors do not discuss hyperparameter optimization for the smaller training sets. Were these fixed for all training sets? Does that then not bias the conclusions?

Answer: We thank the referee for raising this concern. The hyperparameters were all the same for all cases. We initially thought that optimizing the hyperparameters for each case individually could also bias the results (since optimization is done manually), but we agree that using a general model for all cases can also introduce a bias. For that reason, we decided to remove these results from the manuscript.

4. Given the large number of small molecules in the dataset it is very likely that test sets (like CM1 and CM3) contain molecules that are also in the training set. If so this is a problem as we can expect overly optimistic results when evaluating generalization performance.

Answer: The referee is right that this could influence the results. However, CM1 to 4 only have 4, 2, 3 and 4 molecules, respectively, that were included in the training set. We do not believe that, compared to size of these CMs, this has impacted the results given also the fact that the RT prediction for molecules included in the training set is also far from perfect (Figure 1b). We have clarified it in the revised version.

Minor concerns:

1. The authors use statistical tests to address the significance of the small differences they observe. With the sample sizes used in the tests even the smallest difference becomes significant. I think the authors should state more clearly that even though the tests show significant differences, these differences are indeed very small.

Answer: We thank the referee for pointing this out. The new results using a naïve k -NN approach (as suggested by Referee 2) now include a discussion on the small differences between the k -NN and the deep learning approach. We believe that this makes our conclusions stronger as these small differences demonstrate a strong dependency of the structural similarity in the RT prediction.

2. The authors state that R2 values cannot be compared between datasets with different sizes. Why is that?

Answer: We have clarified this point. This is because an R2 has a strong dependency on the size of the dataset and the dataset itself and can not reflect the accuracy differences between two methods. Therefore, when comparing different papers, each paper used its own dataset with different sizes that yield different R2 (influenced by the size of the dataset and also outliers) which in turn biases comparisons between studies/methods. The most reliable variable to compare the performance of RT prediction models is to use the relative and absolute mean or median values that not only account for differences between model sizes but can also be easily interpreted. For example, is a 0.85 vs a 0.89 a big difference? This is difficult to judge for two different datasets, but a difference between a 5% vs a 9% error (or in seconds) is, in our opinion, more interpretable.

REVIEWERS' COMMENTS:

Reviewer #1 (Remarks to the Author):

The authors provide an improved manuscript based on reviewer comments. Thus, I did not have much more comment to do on the manuscript and it can be published in the present form.

Reviewer #2 (Remarks to the Author):

The authors have appropriately answered my comments and suggestions, and revised the paper with new experiments in line with my suggestions. From my point of view it would deserve to be published.

Juho Rousu

Reviewer #3 (Remarks to the Author):

The authors have addressed all my concerns. I believe the dataset presented in the manuscript will add great value to the community.